# Associations of *MDM2* and *MDM4* Polymorphisms with Early-Stage Breast Cancer

**DOI:** 10.3390/jcm10040866

**Published:** 2021-02-19

**Authors:** Agnė Bartnykaitė, Aistė Savukaitytė, Rasa Ugenskienė, Monika Daukšaitė, Erika Korobeinikova, Jurgita Gudaitienė, Elona Juozaitytė

**Affiliations:** 1Oncology Research Laboratory, Oncology Institute, Lithuanian University of Health Sciences, LT-50161 Kaunas, Lithuania; aiste.savukaityte@lsmuni.lt (A.S.); rasa.ugenskiene@lsmuni.lt (R.U.); monika.dauksaite@stud.lsmu.lt (M.D.); 2Department of Genetics and Molecular Medicine, Hospital of Lithuanian University of Health Sciences Kaunas Clinics, LT-50161 Kaunas, Lithuania; 3Department of Oncology and Hematology, Hospital of Lithuanian University of Health Sciences Kaunas Clinics, LT-50161 Kaunas, Lithuania; erika.korobeinikova@lsmuni.lt (E.K.); jurgita.gudaitiene@lsmuni.lt (J.G.); elona.juozaityte@lsmuni.lt (E.J.)

**Keywords:** breast cancer, SNP, *MDM2*, *MDM4*, associations

## Abstract

Breast cancer is one of the most common cancers worldwide. Single nucleotide polymorphisms (SNPs) in *MDM2* and *MDM4* have been associated with various cancers. However, the influence on clinical characteristics of breast cancer has not been sufficiently investigated yet. Thus, this study aimed to investigate the relationship between SNPs in *MDM2* (rs2279744, rs937283, rs937282) and *MDM4* (rs1380576, rs4245739) and I–II stage breast cancer. For analysis, the genomic DNA was extracted from 100 unrelated women peripheral blood. Polymorphisms were analyzed with polymerase chain reaction-restriction fragment length polymorphism (PCR-RFLP) assay. The study showed that *MDM2* rs937283 and rs937282 were significantly associated with estrogen receptor status and human epidermal growth factor receptor 2 (HER2) status. SNPs rs1380576 and rs4245739, located in *MDM4*, were significantly associated with status of estrogen and progesterone receptors. Our findings suggest that rs937283 AG, rs937282 CG, rs1380576 CC, and rs4245739 AA genotypes were linked to hormonal receptor positive breast cancer and may be useful genetic markers for disease assessment.

## 1. Introduction

Breast cancer is one of the most common cancers among women [1,2]. Today many studies claim that breast cancer is a multifactorial disease, and the etiology of cancer is often unknown [3]. However, it has been shown that the major cause is the combination of genetic and environmental factors [1,4,5,6]. It has been demonstrated that overexpression or amplification of *MDM2* and *MDM4* genes are common in many malignancies, including breast cancer [7,8,9,10]. *MDM2*, which is mapped to chromosome 12q14.3–q15, and *MDM4*, which is located on chromosome 1 region q32, encode *MDM2* and *MDM4* proteins, respectively [11]. The evidence suggests that *MDM2* and *MDM4* may play significant roles in breast cancer formation, progression, prognosis, and protection from cancer [7,8,12,13,14]. The cellular processes could be related with other important protein p53, product of *TP53* gene, which is a key regulator in genomic stability, cell cycle, autophagy, apoptosis, and necrosis [4,10,15]. It is known that *MDM2* and *MDM4* proteins perform distinct but cooperative functions in regulation of cellular p53 activity through a combination of p53 degradation and direct transcriptional squelching [16,17,18,19]. TP53 is the most common inactivated tumor suppressor gene in various human cancer types (including breast cancer). In spite of this fact, protein p53 encoded by wild-type *TP53* can be functionally inactivated by abnormal structure or elevated levels of *MDM2* and *MDM4* [9,20,21,22,23]. Consequently, blocking *MDM2* and *MDM4* has been proposed as a cancer treatment strategy [24,25,26,27,28,29,30].

Several studies observed the associations between single nucleotide polymorphisms (SNPs) in *MDM2* and *MDM4* genes and various cancers [9,16,23,24,31,32,33,34,35,36]. In spite of this, the influence on breast cancer has not been sufficiently investigated yet. There is the lack of studies investigating the *MDM2* and *MDM4* associations with the clinical and morphological characteristics such as tumor size, nodal status, histologic grade, estrogen receptor (ER), progesterone receptor (PR), and human epidermal growth factor receptor 2 (HER2) status, proliferation rate, etc. This information might assist as prognostic factor and/or predictive marker for response to treatment of breast cancer. The prognostic and predictive strength of various characteristics is different. However, it was found that women with ER-positive breast cancer, compared to ER-negative, have a better prognosis and the treatment for positive status tumor is more efficacious. Low histologic grade as well as PR-positive and HER2-negative status is also usually associated with better breast cancer prognosis. Meanwhile, high histologic grade, high proliferation rate, and triple negative breast cancer subtype are characterized by unfavorable prognosis [37,38,39,40].

In the present study we determined the relationship between *MDM2* and *MDM4* SNPs and characteristics of breast tumor. The information may improve the assessment of prognostic and/or predictive value of *MDM2* and *MDM4* genotypes in breast cancer patients.

## 2. Materials and Methods

### 2.1. Patients

The homogeneous study group consisted of 100 unrelated Lithuanian women with a diagnosis of primary breast cancer. All women were treated in the Hospital of Lithuanian University of Health Sciences Kaunas Clinics. The age between 30 and 50 years at the time of diagnosis, early stage (I–II) of the disease and premenopausal status were preferred. All clinical and tumor pathomorphological data of the patients were obtained from the medical records with the help of the oncologists. The exclusion criteria were other malignancies, significant comorbidities, and incomplete medical documentation. Study data included the age at diagnosis, pathological tumor size (pT), status of pathological lymph node involvement (N), status of estrogen (ER) and progesterone (PR) receptors, human epidermal growth factor receptor 2 (HER2) status, tumor grade (G1 and G2, G3), progress, metastasis, and death.

The study was performed at the Oncology Research Laboratory (Oncology Institute, Lithuanian University of Health Sciences). The study was approved by Kaunas Regional Biomedical Research Ethical Committee (protocols Nr. BE-2-10 and Nr. P1-BE-2-10/2014). A written informed consent was obtained from all the participants.

### 2.2. DNA Isolation and Genotyping

The blood samples were collected in EDTA-containing tubes from all included subjects in 2014–2017. For SNP analysis genomic DNA was extracted from peripheral blood leukocytes with a commercially available DNA extraction kit (Thermo Fisher Scientific Baltics, Lithuania). The SNPs in *MDM2* and *MDM4* genes were analyzed with polymerase chain reaction-restriction fragment length polymorphism (PCR-RFLP) assay according to a self-made protocol. PCR reactions were carried out in a total volume of 25 μL containing distilled water (dH_2_O), 1x DreamTaq buffer, 0.2 mM of each dNTP, DMSO, 0.24 pmol/μL of forward and reverse primers, 0.02 U DreamTaq polymerase and template DNA. Only the PCR mix for rs937283 was different and composed of distilled water (dH_2_O), 1x Taq buffer (with NH_4_), 0.2 mM of each dNTP, MgCl_2_, 0.4 pmol/μL of forward and reverse primers, 0.025 U Taq polymerase and template DNA. The negative control was included in each experiment to ensure the accuracy of the amplification. The primer sequences were described previously [4,5,41,42,43]. The thermal conditions and primer sequences are shown in Table 1.

DNA sequences from the amplified region of genotypes were screened manually for specific restriction enzyme sites (http://nc2.neb.com/NEBcutter2/ and http://www.restrictionmapper.org/ programs were used). Enzymes with predicted exclusive cutting sites in each genotype were selected and used in RFLP development and analysis (Table 2). All PCR and RFLP products were fractioned electrophoretically (5 V/cm) on a 2 or 3% agarose gel with ethidium bromide.

### 2.3. Statistical Analysis

Statistical analysis was performed by using SPSS (Statistical Package for the Social Sciences) version 20.0 statistical software (SPSS Inc., Chicago, IL, USA). The deviation from Hardy–Weinberg equilibrium was tested (http://ihg.gsf.de/cgi-bin/hw/hwa1.pl). Pearson’s Chi-square or Monte Carlo tests were used to determine the statistical significance of the association between categorical values for each genotype group (genotype model). The odds ratios (ORs) and 95% confidence intervals (CIs) were obtained from univariate logistic regression analyses to evaluate associations between SNPs and cancer characteristics. Additionally, multivariate logistic regression analyses were performed to estimate the adjusted ORs. The multivariate analyses were conducted in three models (No. 1, No. 2 and No. 3). Model No. 1 included combination of SNPs in MDM2 or *MDM4* as potential covariates. In model No. 2, the receptors as additional confounding variables were included. In model No. 3, SNPs and all analyzed breast cancer characteristics (age, status of receptors, pT, N, and G) were considered as potential covariates. The overall survival was measured from the date of diagnosis till the event—last follow-up or death. Survival plots were generated using the Kaplan–Meier method. The differences between genotypes were assessed using the log-rank test. *p* < 0.05 was determined as criterion for statistical significance for all executed statistical tests.

## 3. Results

### 3.1. The Distribution of the Breast Tumor Features, Alleles and Genotypes

Our analysis included 100 Lithuanian breast cancer patients and 65 of them were diagnosed over 40 years. 68% of patients had >1 mm but ≤5 mm tumors and 45% of the subjects had tumors spread to the lymph nodes. 57% of patients had positive ER expression, 48% had positive PR expression, and 22% had increased HER2 expression. In 71% of the subjects, the tumor was well or moderately differentiated (grade G1 and G2). In 31 of 100 cases, cancer had progressed, and 22% women died of breast cancer (Table 3).

Five SNPs across *MDM2* (rs2279744, rs937283, rs937282) and *MDM4* (rs1380576, rs4245739) genes were evaluated for associations with breast cancer phenotype. All observed genotype distributions, excluding *MDM4* rs1380576 (*p* = 0.02), were in agreement with the Hardy–Weinberg equilibrium (*p* > 0.05). The allele and genotype distributions of the studied SNPs are summarized in Table 4.

### 3.2. The Associations of *MDM2* Polymorphisms with Breast Cancer Characteristics

The results of the analysis showed some significant associations between the *MDM2* polymorphisms and tumor features. The SNPs rs937283 and rs937282 were significantly associated with ER (*p* = 0.023 and *p* = 0.021, respectively) and HER2 (*p* = 0.010 and *p* = 0.033, respectively) status. Meanwhile, no significant differences were observed in the association analysis between the rs2279744 genotypes and tumor features (*p* > 0.05). Results are summarized in Appendix A. Furthermore, the effect of SNPs in *MDM2* on the overall survival was analyzed but no significant associations were detected (Appendix A).

As shown in Table 5, after identifying *MDM2* rs937283, it was found that patients carrying the AG genotype were predisposed to higher rates of ER-positive disease (OR = 2.538, 95% CI 1.336–4.823, *p* = 0.004). When compared with the AA genotype, the AG was also significantly associated with decreased chances of HER2-positive breast cancer (OR = 0.231, 95% CI 0.060–0.894, *p* = 0.034). In addition, the logistic regression analysis revealed a significant association (OR = 2.538, 95 % CI 1.366–4.823, *p* = 0.004) between rs937282 CG genotype and ER-positive status (Table 5.)

We also performed a combined evaluation of rs937283 and rs937282 with status of ER and HER2 (Appendix A). However, the analysis of a possible joint effect did not reveal significant results. Following the adjustment for more confounding variables (models No. 2 and No. 3), the associations also remained non-significant.

### 3.3. The Associations of *MDM4* Polymorphisms with Breast Cancer Characteristics

Both SNPs, located in *MDM4*, rs1380576 and rs4245739, showed statistically significant associations with ER (*p* = 0.005 and *p* = 0.010, respectively) and PR (*p* = 0.010 and *p* = 0.016, respectively) status. Results are summarized in Appendix A. Results also showed that SNPs in *MDM4* were not statistically associated with overall survival (Appendix A).

Individuals having the rs1380576 CC genotype had an OR of 2.263 (95% CI 1.319–3.883, *p* = 0.003) of an ER-positive status compared to individual having the GG genotype. Moreover, CC genotype was significantly associated with the PR-positive disease (OR = 12.462, 95% CI 1.486–104.514, *p* = 0.020) (Table 6).

As indicated in Table 6, we found that rs4245739 AA genotype was significantly associated with ER-positive breast cancer in comparison with CC genotype (OR = 1.913, 95% CI 1.155–3.168, *p* = 0.012). After logistic regression analysis, no reliable differences were found between rs4245739 polymorphism genotype and PR-positive status (*p* > 0.05).

However, according to results of multivariate logistic regression analysis, all investigated associations resulted in loss of significance (Appendix A). The results suggested that in this case other factors were more important.

## 4. Discussion

Breast cancer is one of the most common cancers affecting females worldwide [2]. The etiology and pathogenesis of breast cancer is complicated [6]. Various risk factors, including genetic predisposition, have been characterized, but the accurate molecular mechanisms of breast cancer cause and characteristics are still unclear [23]. Additionally, some studies predicate that overexpression/amplification of *MDM2* and *MDM4* are common in many malignancies [7,8,9,17]. Genetic variations (for example, SNPs) in the *MDM2* and *MDM4* genes may also be associated with breast cancer. Moreover, it is established that TP53 pathway is one of the central pathways involved in various human cancers, including breast cancer [23,44,45]. Many studies have shown that *MDM2* and *MDM4* proteins are also involved in TP53 pathway by performing cooperative functions in negatively regulating p53 protein, which is the key regulator in essential cellular processes [16,17,19].

In our study we investigated the *MDM2* (rs2279744, rs937283, rs937282) and *MDM4* (rs1380576, rs4245739) single nucleotide polymorphisms in 100 Lithuanian breast cancer patients. We investigated the relationship of *MDM2* and *MDM4* polymorphisms genotype with the clinicopathological features, including the patient’s age at diagnosis, pathological tumor size (pT), pathological lymph node involvement (N), status of estrogen (ER), progesterone (PR) receptors and human epidermal growth factor receptor 2 (HER2), tumor grade (G1 and G2, G3), progress, metastasis, and death. For comparisons of all genotype and associated features, logistic regression analysis was used to determine odds ratios and *p* values. Our results revealed that some SNPs of *MDM2* and *MDM4* might be significantly associated with the breast cancer characteristics. To the best of our knowledge, this is the first study to examine the role of *MDM2* and *MDM4* polymorphisms in breast cancer patients in Lithuania.

As the data of our study shows, associations for *MDM2* and *MDM4* SNPs with age at breast cancer diagnosis, pathological tumor size, lymph node involvement status, histological grade, progression, metastasis, and death were evaluated but no significant associations were found. In addition, no significant associations between SNPs and overall survival were detected.

*MDM2* rs2279744 polymorphism is located in the promoter region. It has been shown to increase the affinity of the stimulatory protein 1 (Sp1, transcriptional activator), resulting in higher levels of *MDM2* mRNA and protein [46]. In our study, rs2279744 polymorphism did not show any significant association with analyzed breast cancer characteristics. Like our study, Miedl et al. [47] have reported no significant associations of rs2279744 with breast cancer features (age, menopausal status, ER, PR, and HER2 status, stage, grade, and tumor size). Similarly, Yilmaz et al. [48] did not find any relationships among tumor grade, tumor size, lymph node involvement, metastasis status, and this polymorphism. In contrast, Yadav with colleagues [23] have found a significant correlation between rs2279744 and HER2/neu-positive status and distant metastasis (*p* = 0.003 and *p* = 0.04, respectively) in breast cancer. Furthermore, analyzing control and breast cancer cohorts, Paulin et al. [36] indicated that rs2279744 was associated with tumor grade and nodal involvement. Their study revealed that GG genotype was associated with high grade tumors (OR = 1.64, 95% CI = 1.06–2.53, *p* = 0.025) and greater nodal involvement (OR = 2.51, 95% CI = 1.26–4.98, *p* = 0.009). Moreover, in concordance with our findings, they report that rs2279744 was not associated with age at diagnosis of breast cancer, tumor ER, PR, HER2 status, and menopausal status (*p* > 0.05).

rs937283 is located in the 5′ untranslated region (UTR). This genetic variant enhances the transcription activity of the *MDM2* gene thereby increases the mRNA and protein expression levels of *MDM2* [35]. Meanwhile, rs937282 is located in the promoter region of *MDM2*. Similarly to rs2279744, rs937282 can affect the affinity of CAAT/enhancer binding protein α (C/EBP α), resulting in higher expression of *MDM2* [33]. In the present study, we found the relationships between these two SNPs of *MDM2* (rs937283 and rs937282) and ER, HER2 status. We observed that heterozygous genotypes were significantly associated with ER-positive breast cancer. The analysis also revealed a statistically significant association between the rs937283 AG genotype and decreased chances of HER2-positive breast cancer. It is known that ER receptor positive breast cancer is normally associated with much better prognosis. Consequently, our results suggest that heterozygous genotypes of rs937283 and rs937282 polymorphisms could indicate better prognosis for breast cancer patients. Literature sources presented only the data of these SNPs and their influence on the risk of various cancers, but there was no evidence that rs937283 and rs937282 had been analyzed with breast cancer characteristics. Some correlations with other types of tumors have been recently identified. For example, Jiao et al. [35] studied association of rs937283 with clinicopathological characteristics (gender, age at diagnosis, family history, invasion, aggression, and lag-time) in retinoblastoma patients, and significant associations with tumor invasion and high aggression were identified.

*MDM4* rs1380576 is the intronic variant and its exact functional changes remain unknown. We have observed that rs1380576 polymorphism was significantly associated with ER and PR status. In the present study, we found that patients with CC genotype were linked to positive status of ER and PR. Conversely, Hashemi et al. [5] did not show significant associations between rs1380576 and status of ER and PR (*p* > 0.05). In agreement with our study, they have also reported no significant relationship between rs1380576 and other clinical breast cancer characteristics such as age, tumor size, grade, stage, histological type, HER2 status. There is the lack of studies investigating the rs1380576 associations with the clinical and morphological characteristics of breast cancer. Consequently, further studies concerning the role of this SNP may be considered.

rs4245739 polymorphism is located in the 3′ untranslated region (UTR) of the *MDM4* gene, affects mRNA stability and proteins level [31]. In this study, we confirmed the impact of *MDM4* rs4245739 on ER status, and we consider that AA genotype could indicate better breast cancer prognosis. Our findings for rs4245739 and ER status concur with those of Bauer et al. [17]. They analyzed rs4245739 in 815 breast cancer patients and results showed that AA genotype was associated with a 1.8-fold increased probability to develop an ER-positive tumor (*p* = 0.042). We discovered AA genotype to be associated with ER-positive status with OR = 1.913. What is more, according to Purrington et al. [49], rs4245739 C allele was related to ER-negative status (OR = 1.14, 95 % CI 1.10–1.18, *p* = 2.1 × 10^−12^). Meanwhile, Milne et al. [50] in their study claimed that rs4245739 was not associated with ER-positive disease. Moreover, Pedram et al. [51] (Azerbaijan population) and Hashemi et al. [5] (Southeast Iranian population) found that *MDM4* rs4245739 had no significant association with breast cancer clinicopathological factors, such as age, involved breast side, tumor size and type, grade of tumor, count of involved lymph nodes and stage of cancer, status of receptors. Liu with colleagues [4] reported no associations with age, age at menarche, menstrual history, or family history of breast cancer.

We have to admit that this is the first analysis of polymorphisms in the *MDM2* and *MDM4* genes in relation to the breast cancer clinical characteristics in Lithuania; consequently, a much larger cohort of breast cancer patients would be required to verify our findings.

## 5. Conclusions

This study provides basic information about the genotype frequency distributions of *MDM2* rs2279744, rs937283, rs937282 and *MDM4* rs1380576, rs4245739 polymorphisms in Lithuanian population. Our results suggest that heterozygous genotypes of rs937283 and rs937282, as well as rs1380576 CC and rs4245739 AA genotypes, are associated with early-stage hormonal receptor positive breast cancer and may be useful genetic markers for disease assessment. This information may improve patient stratification in the future; however, further large and functional studies are needed to assess the validity of these associations.

## Figures and Tables

**Table 1 jcm-10-00866-t001:** The SNP and thermal conditions for analysis.

Gene, SNP	Primer Sequence F-Forward, R-Reverse	Thermal Conditions	Fragments, bp
*MDM2* rs2279744	F 5′-CGCGGGAGTTCAGGGTAAAG-3′ R 5′-CTGAGTCAACCTGCCCACTG-3′	5 min of denaturation at 95 °C, then 35 cycles of 95 °C for 30 s, 63 °C for 30 s, and 72 °C for 30 s. The final cycle had a 7 min extension at 72 °C	157
*MDM2* rs937283	F 5′-TGACCGAGATCCTGCTGCTTTC-3′ R 5′-TGAGTCAACCTGCCCACTGAAC-3′	5 min of denaturation at 95 °C, then 35 cycles of 95 °C for 30 s, 60 °C for 30 s, and 72 °C for 30 s. The final cycle had a 7 min extension at 72 °C	617
*MDM2* rs937282	F 5′-GGTAACAGCGACACGGAGAT -3′ R 5′-CTCCGGGATGATGGAGTG -3′	5 min of denaturation at 95 °C, then 35 cycles of 95 °C for 30 s, 57,5 °C for 30 s, and 72 °C for 30 s. The final cycle had a 7 min extension at 72 °C	231
*MDM4* rs1380576	F 5′-GAAGAGGTGACATTTAACCTGAGACTT-3′ R 5′-GTGGTCTATCCCCTCAGCACATTTCCA-3′	5 min of denaturation at 94 °C, then 35 cycles of 94 °C for 30 s, 56,6 °C for 45 s, and 72 °C for 30 s. The final cycle had a 10 min extension at 72 °C	195
*MDM4* rs4245739	F 5′-AAGACTAAAGAAGGCTGGGG-3′ R 5′-TTCAAATAATGTGGTAAGTGACC-3′	3 min of denaturation at 94 °C, then 35 cycles of 94 °C for 30 s, 51 °C for 30 s, and 72 °C for 30 s. The final cycle had a 10 min extension at 72 °C	134

Nucleotide base in underlined letter is mismatched base added in the primer sequence to create restriction enzyme site.

**Table 2 jcm-10-00866-t002:** The SNP and restriction conditions for analysis.

Gene and SNP	Restriction Endonuclease	Restriction Conditions	Fragments, bp
*MDM2* rs2279744	MspA1I	37 °C for 1 h	T allele 157; G allele 109 + 48
*MDM2* rs937283	MboI	37 °C for 1–16 h	A allele 437 + 87 + 86 + 7; G allele 524 + 86 + 7
*MDM2* rs937282	Hpy188I	37 °C for 1 h	C allele 231; G allele 182 + 49
*MDM4* rs1380576	BseNI (BsrI)	65 °C for 1–16 h	G allele 195; C allele 172 + 23
*MDM4* rs4245739	MspI (HpaII)	37 °C for 1–16 h	C allele 110 + 24; A allele 134

**Table 3 jcm-10-00866-t003:** The clinicopathological characteristics of the study group.

Characteristics	Frequencies (%)
Age	
30–40 years	35
41–50 years	65
Pathological tumor size (pT)	
T1a	68
T1b	32
Pathological lymph node involvement (N)	
negative (N0)	55
positive (N1)	45
Estrogen receptor (ER)	
negative	43
positive	57
Progesterone receptor (PR)	
negative	52
positive	48
Human epidermal growth factor receptor 2 (HER2)	
negative	78
positive	22
Tumor grade (G)	
G1 and G2	71
G3	29
Progress	
absent	69
present	31
Metastasis	
absent	74
present	26
Death	
absent	78
present	22

T1a—>1 mm but ≤5 mm, T1b—>5 mm but ≤10 mm across, G1—well differentiated (low grade), G2—moderately differentiated (intermediate grade), G3—poorly differentiated.

**Table 4 jcm-10-00866-t004:** Allele and genotype distribution of *MDM2* and *MDM4* polymorphisms in the study group.

Polymorphism	Allele Frequency (European Population Allele Frequencies from 1000 Genomes Project Database)		Genotype Frequency		
*MDM2* rs2279744 NG_016708.1:g.5610T>G	T-0.68 (0.65)	G-0.32 (0.35)	TT-0.49	TG-0.38	GG-0.13
*MDM2* rs937283 NG_016708.1:g.5194A>G	A-0.47 (0.61)	G-0.53 (0.39)	AA-0.24	AG-0.46	GG-0.30
*MDM2* rs937282 NG_016708.1:g.4827C>G	C-0.46 (0.51)	G-0.54 (0.49)	CC-0.23	CG-0.46	GG-0.31
*MDM4* rs1380576 NG_029367.1:g.7772G>C	G-0.24 (0.31)	C-0.76 (0.69)	GG-0.10	GC-0.28	CC-0.62
*MDM4* rs4245739 NG_029367.1:g.38336C>A	C-0.20 (0.26)	A-0.80 (0.74)	CC-0.07	AC-0.26	AA-0.67

**Table 5 jcm-10-00866-t005:** Odds ratio for associations of rs937283 A > G and rs937282 C > G with tumor characteristics.

SNP	Genotype	Positive ER			Positive HER2		
		OR	95% CI	*p*	OR	95% CI	*p*
*MDM2* rs937283	AA	1.00	(ref.)	-	1.00	(ref.)	-
	AG	2.538	1.336–4.823	0.004	0.231	0.060–0.894	0.034
	GG	0.765	0.371–1.574	0.467	1.406	0.444–4.448	0.562
*MDM2* rs937282	CC	1.00	(ref.)	-	1.00	(ref.)	-
	CG	2.538	1.366–4.823	0.004	0.346	0.093–1.287	0.113
	GG	0.722	0.354–1.474	0.371	1.558	0.476–5.104	0.464

**Table 6 jcm-10-00866-t006:** Odds ratio for associations of rs1380576 G>C and rs4245739 C>A with tumor characteristics.

SNP	Genotype	Positive ER			Positive PR		
		OR	95% CI	*p*	OR	95% CI	*p*
*MDM4* rs1380576	GG	1.00	(ref.)	-	1.00	(ref.)	-
	CG	0.647	0.303–1.381	0.261	5.824	0.645–52.599	0.117
	CC	2.263	1.319–3.883	0.003	12.462	1.486–104.514	0.020
*MDM4* rs4245739	CC	1.00	(ref.)	-	1.00	(ref.)	-
	AC	0.857	0.396–1.853	0.695	0.733	0.337-1.597	0.435
	AA	1.913	1.155–3.168	0.012	1.233	0.762–1.996	0.393

## Data Availability

The data presented in this study are available on request from the corresponding author.

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
