# Peer review of "Associations of MDM2 and MDM4 Polymorphisms with Early-Stage Breast Cancer"

_jcm, 2021, doi:10.3390/jcm10040866_

Round 1

Reviewer 1 Report

The manuscript “ Associations of MDM2 and MDM4 polymorphisms with early stage breast cancer” by Bartnykaite et al. analyzed three MDM2 (rs2279744, rs937283, rs937282) and two MDMX (rs1380576, rs4245739) polymorphisms in 100 unrelated breast cancer patients using PCR-RFLP, and further assessed the potential association between these polymorphisms and breast cancer subtypes and clinical characteristics.  Epidemiological and statistical approach is a powerful and efficient way to identify genetic variants that are associated with breast cancer risk factors. As the cellular p53-MDM2-MDM4 regulatory circuit plays a crucial role in breast cancer development and prevention, the analyses of specific MDM2 and MDM4 SNPs and their association with the breast cancer subtypes and prognosis markers could be quite informative. Specific questions and comments are summarized blow. The authors should consider to address the following comments and to improve the quality of the manuscript.

  • The study analyzed 100 breast cancer patients, but did not include any healthy control individual. What are the allele and genotype frequency of the MDM2 and MDM4 SNPs in healthy group? Authors may consider to report or cite MDM2 and MDM4 SNPs allele and genotype frequencies in healthy controls and compare with the current study cohort. This information will serve as a control, help to interpret the association data between MDM2 and MDM4 SNPs and some of breast cancer variants included in this study (age, disease progress etc).
  • As for the method used for SNP genotyping, is there any exclusions or any samples that could not be genotyped for analysis? Have the genotyping results been validated with any other methods?
  • The current study examined the potential association between breast cancer subtypes and each MDM2 and MDM4 SNPs. Is there any joint effect with combinations of these SNPs on breast cancer characteristics? Potential cofounding effects between different SNPs and their association with breast cancer subtypes could be further assessed.

Author Response

Point 1: English language and style. English language and style are fine/minor spell check required

Response 1: We appreciate the positive feedback from the Reviewer.

Point 2: Does the introduction provide sufficient background and include all relevant references? Yes

Response 2: We appreciate the positive feedback from the Reviewer.

Point 3: Is the research design appropriate? Can be improved

Response 3: We have carried out additional statistical analysis and hope that this has improved the design and quality of the study. Specific changes have been discussed in response to points 8 and 10.

Point 4: Are the methods adequately described? Yes

Response 4: We appreciate the positive feedback from the Reviewer.

Point 5: Are the results clearly presented? Yes

Response 5: We appreciate the positive feedback from the Reviewer.

Point 6: Are the conclusions supported by the results? Can be improved

Response 6: We have contributed our best efforts to improve the conclusions. Please see lines 31-33 on page 1 and 342-352 on page 10.

Point 7: The manuscript “ Associations of MDM2 and MDM4 polymorphisms with early stage breast cancer” by Bartnykaite et al. analyzed three MDM2 (rs2279744, rs937283, rs937282) and two MDMX (rs1380576, rs4245739) polymorphisms in 100 unrelated breast cancer patients using PCR-RFLP, and further assessed the potential association between these polymorphisms and breast cancer subtypes and clinical characteristics. Epidemiological and statistical approach is a powerful and efficient way to identify genetic variants that are associated with breast cancer risk factors. As the cellular p53-MDM2-MDM4 regulatory circuit plays a crucial role in breast cancer development and prevention, the analyses of specific MDM2 and MDM4 SNPs and their association with the breast cancer subtypes and prognosis markers could be quite informative. Specific questions and comments are summarized blow. The authors should consider to address the following comments and to improve the quality of the manuscript.

Response 7: We appreciate the positive feedback from the Reviewer. We have answered each of his/her points below.

Point 8: The study analyzed 100 breast cancer patients, but did not include any healthy control individual. What are the allele and genotype frequency of the MDM2 and MDM4 SNPs in healthy group? Authors may consider to report or cite MDM2 and MDM4 SNPs allele and genotype frequencies in healthy controls and compare with the current study cohort. This information will serve as a control, help to interpret the association data between MDM2 and MDM4 SNPs and some of breast cancer variants included in this study (age, disease progress etc).

Response 8: This study was designed for the disease assessment. We analyzed the impact of single nucleotide polymorphisms (SNPs) in MDM2 and MDM4 genes on breast cancer morphology and course of disease. The aim of the study was to determine the relationship between SNPs and clinical characteristics of breast tumor and prognosis, not to investigate the associations of SNPs with the risk of breast cancer development. Thus, the prognostic role of polymorphisms have been evaluated and a healthy control group was not required for this purpose. However, following the Reviewer‘ comments, the information concerning allele frequencies according to European population data from 1000 Genomes Project database was added in the Table 4.

Point 9: As for the method used for SNP genotyping, is there any exclusions or any samples that could not be genotyped for analysis? Have the genotyping results been validated with any other methods?

Response 9: Blood samples were collected in EDTA-containing vacutainer tubes from 100 selected breast cancer patients (the exclusion criteria were applied for the patients before samples collection and were noted in the article). All collected samples were good quality, suitable for DNA extraction and further analysis. After PCR and RFLP protocols optimization all samples were amplified and restricted. Following the first run of the samples for each SNP analysis, the repressenting samples of all genotypes were run with other samples as a “positive control“. In all the cases the genotype of the “positive control“ was in aggreement with the previous run, thus all 100 samples were genotyped successfully. Since results of genotyping with PCR-RFLP assay did not cause any inconveniences, other methods have not been used.

Point 10: The current study examined the potential association between breast cancer subtypes and each MDM2 and MDM4 SNPs. Is there any joint effect with combinations of these SNPs on breast cancer characteristics? Potential cofounding effects between different SNPs and their association with breast cancer subtypes could be further assessed.

Response 10: We have carried out additional multivariate logistic regression analysis. More information about it can be found in the statistical analysis section (lines 124-130 on page 4). The results are presented in additional file: Table 2S and lines 177-180 on page 6 and 208-210 on page 7. Additionally, we reviewed the study and removed the information and results concerning the intrinsic subtypes. We decided that this is the unnecessary redundant data which did not provide additional important information to our study.

Reviewer 2 Report

The authors have examined polymorphisms at 5 loci in genes coding for the p53 regulatory proteins MDM2 and MDM4 in a small cohort of female breast cancer patients in Lithuania. The results show some significant associations between several SNPs and standard histological biomarkers for determining subtype, predominantly ER expression from which most of the other characteristics derive. The authors suggest a utility for their findings in prediction and prognostication, but the findings do not add any significant value in this regard beyond what is readily available from standard clinical assays. With the known functional consequences of the SNPs in terms of protein expression and established roles for MDM2/4 in both normal breast development and breast cancer biology, the findings are more interesting from the viewpoint of tumour aetiology.

Discussion – the results of this study appear to be at odds with a significant body of the literature. A statement is made in relation to a single study to suggest that this discrepancy may be due to different allele frequencies in different populations (which doesn’t really make sense and suggests that other factors are far more important that these polymorphisms). A more thoughtful discussion of the discrepancies is warranted, particularly where several other studies support each other in disagreeing with the current study.

Conclusions – The data presented do not really suggest this at all. There is an association with ER positivity from which a better prognosis could be inferred, but this association is not complete and is therefore much better represented by simply assessing ER status – a standard, routine clinical test.

Minor issues:

Line 155, 173, 176 The way the results are written here is correct in the context but forces the reader to constantly remind themselves that this is a cohort in which breast cancer has already developed. For example, a statement such as “Individuals having the rs1380576 CC genotype had an OR of 2.263 (95 % CI 1.319-173 3.883, p=0.003) for developing ER-positive breast cancer compared to individual having the GG genotype” means something very different if applied to the general population. It would be helpful if these statements could be reframed slightly to help guide the busy reader e.g. “Individuals having the rs1380576 CC genotype had an OR of 2.263 (95 % CI 1.319-173 3.883, Exp=0.003) of an ER-positive status compared to individual having the GG genotype”. Line 153 does this more clearly, for example.

Patients –the authors should provide more information on the patient cohort e.g. how many centres were recruited, what geographical area is covered, over what time period were patients recruited, how were patients selected e.g. was it the first 100 patients through the door that matched the criteria – the methods state a preference for younger patients (30-50), how was this implemented?

Line 54-55 – references all associated with MDM4 whereas MDM2 inhibition has gained much more clinical traction to date (clinical trials for idasanutlin, CGM097, HDM201 and others)

Line 201 – this statement is misleading, a relationship was investigated, not demonstrated for most of these factors.

Author Response

Point 1: English language and style. English language and style are fine/minor spell check required

Response 1: We appreciate the positive feedback from the Reviewer.

Point 2: Does the introduction provide sufficient background and include all relevant references? Can be improved

Response 2: According to point 12 the additional relevant references were added in the introduction section (line 57 on page 2).

Point 3: Is the research design appropriate? Yes

Response 3: We appreciate the positive feedback from the Reviewer.

Point 4: Are the methods adequately described? Can be improved

Response 4: Following the Reviewer‘ comments, the additional information has been written in the section of materials and methods. This has been discussed in response to point 11.

Point 5: Are the results clearly presented? Can be improved

Response 5: We have contributed our best efforts to improve the presentation of results and this has been discussed in response to point 10. Additionally, following the other Reviewers’ comments, we have carried out additional statistical analysis (lines 124-130 on page 4). The results are presented in additional file: Table 2S and lines 177-180 on page 6 and 208-210 on page 7.

Point 6: Are the conclusions supported by the results? Must be improved

Response 6: The conclusion has been rewritten based on the results. Please see lines 31-33 on page 1 and 342-352 on page 10.

Point 7: The authors have examined polymorphisms at 5 loci in genes coding for the p53 regulatory proteins MDM2 and MDM4 in a small cohort of female breast cancer patients in Lithuania. The results show some significant associations between several SNPs and standard histological biomarkers for determining subtype, predominantly ER expression from which most of the other characteristics derive. The authors suggest a utility for their findings in prediction and prognostication, but the findings do not add any significant value in this regard beyond what is readily available from standard clinical assays. With the known functional consequences of the SNPs in terms of protein expression and established roles for MDM2/4 in both normal breast development and breast cancer biology, the findings are more interesting from the viewpoint of tumour aetiology.

Response 7: We appreciate the feedback from the Reviewer. We have answered each of his/her points below.

Point 8: Discussion – the results of this study appear to be at odds with a significant body of the literature. A statement is made in relation to a single study to suggest that this discrepancy may be due to different allele frequencies in different populations (which doesn’t really make sense and suggests that other factors are far more important that these polymorphisms). A more thoughtful discussion of the discrepancies is warranted, particularly where several other studies support each other in disagreeing with the current study.

Response 8: Only a few studies in other populations have analyzed the association of these SNPs with breast cancer features. Due to the small number of similar researches investigating the MDM2 and MDM4 associations with characteristics of breast cancer in the world (especially, in European population), it was really difficult to compare our results with other studies. However, the discussion has been modified. Please see lines 246-249 on page 8, 288-297 and 335-336 on page 9.

Point 9: Conclusions – The data presented do not really suggest this at all. There is an association with ER positivity from which a better prognosis could be inferred, but this association is not complete and is therefore much better represented by simply assessing ER status – a standard, routine clinical test.

Response 9: We had no purpose to investigate the associations with specificity of routine clinical tests. This paper describes a cohort study that aimed to examine the contribution of polymorphisms in MDM2 (rs2279744, rs937283, rs937282) and MDM4 (rs1380576, rs4245739) genes to the clinicopathologic features of breast cancer. Our findings confirmed the hypothesis that these SNPs may be as prognostic biomarkers for early‑stage breast cancer in Lithuanian population. We have contributed our best efforts to improve the conclusions. Please see lines 31-33 on page 1 and 342-352 on page 10.

Point 10: Minor issues: Line 155, 173, 176 The way the results are written here is correct in the context but forces the reader to constantly remind themselves that this is a cohort in which breast cancer has already developed. For example, a statement such as “Individuals having the rs1380576 CC genotype had an OR of 2.263 (95 % CI 1.319-173 3.883, p=0.003) for developing ER-positive breast cancer compared to individual having the GG genotype” means something very different if applied to the general population. It would be helpful if these statements could be reframed slightly to help guide the busy reader e.g. “Individuals having the rs1380576 CC genotype had an OR of 2.263 (95 % CI 1.319-173 3.883, Exp=0.003) of an ER-positive status compared to individual having the GG genotype”. Line 153 does this more clearly, for example.

Response 10: We apologize for not making the results clear enough. We have revised the results and some sentences were reframed slightly (please see lines 165-167 on page 6, 191-194 and 200-201 on page 7, 269 on page 8, 288 and 321 on page 9). Additionally, we reviewed the study and removed the information and results concerning the intrinsic subtypes. We decided that this is the unnecessary redundant data which did not provide additional important information to our study.

Point 11: Patients –the authors should provide more information on the patient cohort e.g. how many centres were recruited, what geographical area is covered, over what time period were patients recruited, how were patients selected e.g. was it the first 100 patients through the door that matched the criteria – the methods state a preference for younger patients (30-50), how was this implemented?

Response 11: The study consisted of 100 adult Lithuanian women with primary I‑II stage breast cancer. All patients were treated in the Hospital of Lithuanian University of Health Sciences Kaunas Clinics. The majority of women were from Kaunas district. The blood samples were collected in 2014-2017. The study included the patients who were treated during that period, met the criteria (young age at the time of diagnosis, early stage (I–II) of the disease, premenopausal status, complete medical documentation, abscence of other malignancies or significant comorbidities), and agreed to participate in the study. The information about the cohort was added in the section of materials and methods. The additional information is presented in lines 78-81 and 95-96 on page 2.

Point 12: Line 54-55 – references all associated with MDM4 whereas MDM2 inhibition has gained much more clinical traction to date (clinical trials for idasanutlin, CGM097, HDM201 and others)

Response 12: We are grateful for the observations. We have studied the literature and supplemented the list of references (lines 57 on page 2 and 433-441 on page 11).

Point 13: Line 201 – this statement is misleading, a relationship was investigated, not demonstrated for most of these factors.

Response 13: We apologize for the misleading. As suggested by the Reviewer, we have corrected the text. Please see line 226 on page 7.

Round 2

Reviewer 1 Report

The authors have adequately answered and addressed the comments.